# Hospitals' Financial Health in Rural and Urban Areas in Poland: Does It Ensure Sustainability?

**Agnieszka Bem [1], Rafał Siedlecki [1], Paweł Prędkiewicz [1], Patrizia Gazzola [2]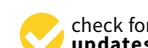, Bożena Ryszawska [1] and Paulina Ucieklak-Jeż [3],***

1   Department of Corporate Finance and Public Finance, Wrocław University of Economics,
    53-345 Wrocław, Poland; agnieszka.bem@ue.wroc.pl (A.B.); rafal.siedlecki@ue.wroc.pl (R.S.);
    pawel.predkiewicz@ue.wroc.pl (P.P.); bozena.ryszawska@ue.wroc.pl (B.R.)
2   Department of Economics, Università degli Studi dell'Insubria, 21100 Varese, Italy;
    patrizia.gazzola@uninsubria.it
3   Faculty of Philology and History, Jan Dlugosz University in Czestochowa, 42-200 Częstochowa, Poland
*   Correspondence: p.ucieklak@o2.pl; Tel.: +48-502-296-808

**Abstract:** Literature review suggests that rural hospitals are in the worst financial conditions due to several factors: They are smaller, located in remote areas, and they provide less specialized services due to their problems with employing well-qualified staff. We decided to check whether it is true in the case of Polish hospitals. Based on the literature review, we have assumed that rural hospitals have less favorable financial conditions. In order to verify this assumption, we use seven indicators of financial health as well as a synthetic measure of financial condition. We have found that, in fact, there is no difference in financial condition between rural and urban hospitals, or even that the financial health of rural hospitals is better if we employ the synthetic measure. Additionally, we have found that the form of activity can be a crucial driver of better financial performance. The concept of rural sustainability is supported by good financial conditions of rural hospitals, which helps to provide better access to medical services for inhabitants of rural areas.

**Keywords:** hospital; rural and urban hospitals; healthcare; sustainable rural health; the financial condition

## 1. Introduction

Sustainability is a model of an economy based on increased social and environmental responsibility [1]. The environmental dimension means introducing low-carbon green economy, which decouples economic growth from consumption of natural resources and energy, at the same time reducing the pressure on the planet by lower emission of $CO_2$ and energy, and resource efficiency. The social dimension supports the idea of responsible consumption, social justice, and equality (both inter- and intra-generational). Transition to sustainability is a long-term, multi-level, complex, and holistic process which involves many actors/stakeholders. The concept of sustainability is tightly bound with technology changes, innovations, and the general digital revolution. Sustainability transition is creating an alternative model of the economy, setting new priorities in social development, and inspiring radical change of attitude towards natural environment, climate, and energy issues [2].

There is a growing recognition that achieving sustainability rests almost entirely on achieving the balance between economic, social, and environmental aspects of development. It also emphasizes the crucial role of the social aspect. It is a large-scale societal transition made by many factors as an agent of change, and the specificity of the concept is associated with public involvement, activism, participation, and has a holistic character [3]. This is interesting from the point of view of policymakers because they have to deal with many dimensions of the crisis and the idea of sustainable development

is offering the path to transform the economy, but also to support citizens and the environment [4] to achieve a higher quality of life [5]. The problems associated with achieving sustainability of the system affect particularly rural areas, which are struggling with deficits in almost every area [6].

While health is one of the most important fields of social and environmental change, fair access to health benefits is one of the most important goals of the health system. According to that, Fineberg [7] suggests new additional attributes which should characterize a sustainable healthcare system: Affordability, acceptability, and adaptability. It is relatively difficult to ensure, mostly due to high information asymmetry [8] and fast technological progress [9,10], or an innovative approach to health care services [11]. The other hidden factor is health communication, which has the strongest impact on people's positive perceptions about healthcare quality [12,13]. Regardless of the above factors, rural areas, by nature, are less equipped with the health care infrastructure, which is the result of sparser population density. This means reduced potential access to benefits—patients living in rural areas, economically more fragile than residents of urban areas, must usually overcome a longer distance to a doctor or a hospital, which requires transportation as well as engages time and financial resources. Hence, the closure of rural healthcare providers deteriorates the availability of benefits.

In this study we focus on the hospital sector due to several reasons: It consumes an important part of financial resources, offers life-saving benefits, and is characterized by high fixed costs. We make an assumption, that a better financial condition is crucial from the point of view of the continued existence of a hospital, as well as the quality of provided services. Poor financial conditions force a change of scope of a hospital's activity, its commercialization, or even closure. According to that, the aim of our research is to assess and compare the financial conditions of rural and urban hospitals in order to examine whether rural hospitals are at higher risk of financial distress. We take into account the fact that the consequences of a rural hospital's closure are more severe than in the case of hospitals located in large cities. A weaker financial condition of rural hospitals might potentially, not only decrease the access to health benefits, but also lower the quality of the provided services. Due to that, sustainable development of rural areas requires actions which would strengthen rural entities. Based on the literature review and previous studies, we propose the following research hypothesis *(H1): Rural hospitals are characterized by poorer financial conditions than urban entities*. We assume that an acceptable level of financial performance can maintain the existence of rural hospital infrastructure. If in fact, rural hospitals are at higher risk of financial distress, then they should be a subject of a policy aimed to strengthen their potency to survive.

The paper is organized as follows: After the introduction (Section 1), we briefly present a health care system in Poland, then (Section 2) we describe the role of rural hospitals as an essential part of sustainable rural development. Based on the review of the literature, we indicate factors which cause the poor financial condition of rural hospitals. In Section 3, we describe the design of the presented study. Next sections are devoted to the description of the data (Section 4) and the methodology (Section 5). In Section 6, we present results and discussion which are followed by the conclusions (Section 7).

The data has been obtained from the Emerging Markets Information System (EMIS) Database, covering the years 2012–2016 and our analysis is supported by Statistica 13.1 and Gretl.

## 2. Health Care System in Poland—Brief Description

Health care in Poland is organized on the basis of a system of universal health insurance. Insurance premiums are discharged from all categories of income and the system does not allow the possibility to substitute the mandatory public insurance with a private one. Financial resources are collected and distributed by the monopolist payer—National Health Fund (Narodowy Fundusz Zdrowia, NFZ). One of the main problems is the low level of funding—the current expenditure on health is just 6.52% (last available data for 2016) of GDP (Gross Domestic Product) and 4.55% GDP comes from public sources (generally the public insurance scheme). Patients significantly participate in the financing of health—the out-of-pocket spending is 23% of current healthcare expenditure. The private sector

is strong in the area of outpatient care—a large part of the providers, particularly in primary health care are non-public actors. The hospital care sector is dominated by public entities—hospitals are mainly owned by local government entities for which providing access to inpatient care is one of the statutory tasks.

Hospitals operate in two basic forms—Independent Public Health Care Institution (SPZOZ) or companies; regardless of the form of the activity, all hospitals owned by the public sector are non-profit. Purely private hospitals play a marginal role from the point of view of access to benefits and, as the for the public ones, they are dependent on public funds. The contracts between hospitals and the National Health Fund specify a range of benefits and the amount of them. The healthcare services provided above the predetermined limit are remunerated only partially or not at all, which is often a source of financial distress. From the year 2018, the network of hospital providers was launched, under which hospitals are paid a flat rate. There are two main differences between SPZOZ and companies: Hospitals operating in the form of a company must keep the financial discipline due to the risk of bankruptcy, but on the other hand, they can benefit from additional sources of income by selling services within supplementary private insurance schemes or out-of-pocket spending. Hospitals operating as SPZOZ cannot effectively go bankrupt, but they are forbidden to privately sell the same services like the ones provided within contracts with the public payer.

Most of the hospitals in Poland struggle with financial difficulties. This applies both to the large hospitals in cities as well as to small hospitals located in rural areas. However, while large urban hospitals are supported by wealthier urban local governments, which, as the owners, finance their investments or supplement the shortage of financial resources, the hospitals in rural areas cannot benefit from such support. Poorer local governments do not have the ability to finance such hospitals and are not able to bear the financial responsibility for the debts of their medical entities (due to the public debt limits). As a result, such local governments seek the possibilities of commercializing or even privatizing their hospitals to get rid of this responsibility.

## 3. Rural Hospitals as Part of a Sustainable Health System

Sustainable rural development is crucial to the economic, social, and environmental development of societies. The discussion on sustainable development emerged in the 1970s and 1980s because humanity became aware that production and consumption destroy the foundations of their life through the over-exploitation of natural resources [14]. The concept of sustainable development was used for the first time at the Stockholm Conference in 1972, in connection with the discussion of the tasks and goals of global environmental protection. The transfer of this concept to international documents took place at the UN Conference "Environment and Development" in Rio de Janeiro in June 1992. The famous Burtland World Commission on Environment and Development in 1987 defined sustainable development as development that meets the needs of the present without compromising the ability of future generations to meet their own needs [15]. Existing definitions emphasize three interconnected aspects of the concept: Environmental, social, and economic. Sustainable development means managing the use of natural environmental resources and the organization of social life, which will improve and then preserve a high quality of life [16] maximizing the net benefits of economic development in the long-term perspective [17]. Other authors note that sustainable development means the evolution of human society [18,19].

The idea of sustainable development became an important element of political debate and the priority of development strategies and sectorial policies. It combined the three main pillars of economic and social development with the protection of the environment and its resources. Sustainable development as an international political idea defined the concept of sustainable production and consumption, and sustainable transport. It also encouraged people to mitigate climate change and strongly emphasized the importance of the problems of poverty, inequality, social exclusion, and the public-health sector.

The current evolution of the sustainable development concept under the guidance of the United Nations leads us to the turning point. The United Nations (UN) Member States in 2015 adopted the 2030 Agenda for Sustainable Development with the 17 sustainable development goals (SDGs). As a result, a new systemic holistic definition of sustainable development was identified with the core principle of global cooperation and national development [20]. Agenda 2030 is a future, long-term social contract for the world. Recently, SDGs became a benchmark for countries, local communities, business, and NGOs, and they are present in many strategies and policies. The core part of the new approach is social sustainability (exclusion, poverty, inequalities) represented by the sentence "no one will be left behind". So, economic and environmental sustainability must be aligned with social sustainability [21].

The rural and urban areas are facing strong global challenges, especially those related to poverty, exclusion, lack of justice, inequality, climate, environmental damage, and peace. SDGs address these issues and formulate a new social, economic, and environmental vision. For example, goal 1: No poverty says that economic growth must be inclusive and promote equality. The next goal proposed is Zero Hunger and it emphasizes the role of rural areas and agriculture sector in providing food, stopping hunger, and eradicating poverty. Very important for quality of life goals 3 and 4 are good health and wellbeing, and quality education—ensuring healthy lives and wellbeing are essential to sustainable development, and obtaining a quality education is the foundation for improving people's lives and sustainable development. Next, the crucial goal for sustainability is to reduce inequalities (Goal 10). It means that policies should be universal in principle, paying attention to the needs of disadvantaged and marginalized populations [22]. The new, multidimensional, complex identification of sustainability is unique because, except all specific goals, it is underlining the importance of cooperation of everybody with the leading role of the government and its institutions in implementing just transition to sustainability [23]. The transformations require governance structures and capabilities, political action, and the formation of actors of change [24].

Therefore, coordinating rural development initiatives that contribute to sustainable social environment is critical. In different parts of rural Europe, a new paradigm of sustainable rural development has begun to take hold. There are several main reasons for the emerging new paradigm: (1) The 'squeeze' on European agriculture, (2) new sources of income, (3) changing role of rural areas from food production to multifunction, and (4) the aesthetic-consumptive functions of places [25]. The environmental aspects of the definition of traditional sustainability mean that agriculture is not only regarded as an economic sector and food-producing sector but also has to maintain multifunctional green space and landscape quality as well [26]. Farmers get direct payment to protect the environment and landscape and develop agro-tourism. On this basis, we define sustainable rural development as territorially based development that redefines nature by re-emphasizing food production and agro-ecology and that re-asserts the socio-environmental role of agriculture as a major agent in sustaining rural economies and cultures [27]. The 'new rural paradigm' (NRP) includes a new, multi-sector, place-based approach to rural development that claims the need for closer cooperation and synergy between the rural and urban economy, and towards rural development as a way to reduce exclusion, poverty, and inequalities on the regional and local level more generally [28]. It presents the emerging tendency of decentralization, re-localization, and self-organization, which is a new regional paradigm, resulting in new linkages among sectors, businesses, producers, consumers, and markets [29].

Sustainable development also means a new perspective on health issues, where sustainable development is perceived as a part of the wider concept consisting of health, wellbeing, economy, environment, and social justice [30]. According to the social pillar of the traditional definition of the sustainable development, one of the important roles of the rural hospital is to help the community. The existence of hospitals in rural areas improves access to health benefits so the rural hospitals' surviving is a key factor of sustainable rural health [31]. It is very important because rural inhabitants are less probable to benefit regularly from doctor's consultation and have weaker access to emergency

services [32], which are usually situated in more urbanized areas. Rural hospitals can leverage their strong relationships with local communities and patients; they have also a significant positive influence on health of the population. When a hospital is closed, patients are forced to seek a new provider, which sometimes means a break in therapy [33]. A remote location also generates costs of transport, if it is available at all. This problem can be particularly important in the case of the elderly or persons suffering from chronic diseases [34]. This prolonged travel time to the hospital also increases mortality in cases of emergency [35].

According to the economic pillar of the traditional definition of the sustainable development, all hospitals operate using valuable and scarce resources and they have faced significant changes over recent decades [36]. Those changes of economic environment resulted in many countries in massive closures of rural hospitals due to their poor financial condition [37–39], so rural hospitals must adapt to reach the level of sustainability [31]. In most countries, hospitals located in rural areas are smaller and deliver less specialized services. Despite this smaller range of services, rural hospitals play an important role in satisfying the rural population's health needs [40–43] or just in being part of local health infrastructure [44], which is so crucial because barriers in access to hospital, or broadly speaking, health services, impacts rural population's health outcomes [45]. Furthermore, we consider health services higher than public and private services. Rural health services have infrastructure and people that are part of local communities. Health professionals are often important members of their communities and they are also actively engaged in the social life of the rural communities.

Rural hospitals impact economic rural development through their health care infrastructure, employment of doctors, dentists, nurses, the quality of medical and health services, and pharmacies. The localization of rural hospitals and medical services encourage localization of care centers for the elderly and influence the development of the silver economy. It means that there is the demand for supply of different services and goods for retired people. In the past two decades, consumer spending among those aged 60 and over increased 50% faster when compared to those under 30. Rural hospital infrastructure increases the attractiveness of a community for physicians as well as for retail business and manufacturing firms. Thus, it might indirectly affect the overall level of community economic activity [46]. Examples of sectors expected to benefit significantly from the silver economy are cosmetics and fashion, tourism, smart homes supporting independent living, service robotics, health (including medical devices, pharmaceuticals, and e-Health) and wellness, safety, culture, education and skills, entertainment, personal and autonomous transport, banking, and relevant financial products [47].

Finally, it increases the growth of sales, consumption, and taxes for local budget. Rural hospitals bring in money from the outside, for example from National Health Funds, and they positively support the prosperity of the local economy [48]. From the economic and social point of view of the traditional definition of sustainability, the health sector provides significant direct benefits through employment and growing incomes. The employment of one physician in a rural area can create an additional five jobs and have an impact on income, retail sales, and sales tax collections. The study concludes that a physician plays a vital role in the economy of the host community. Many researchers in the US confirmed that rural hospitals are the second largest employer in rural counties [46].

Based on the literature review, we can indicate factors contributing to rural hospitals' poor financial condition. They are:

(1) Smaller size [37,49–57], which seems to be the most important determinant of lower profitability and poor financial performance;
(2) Lower elasticity and higher sensitivity to changes [58,59];
(3) Lack of skilled professionals [34,44,60–62];
(4) Poor equipment [61];
(5) A small range of benefits [50,53,55,60,63];
(6) Lower bed occupancy [34,41] declining inpatient admissions [60], the lower economy of scale [57,61].

Those risk factors are generally the consequences of the remote location [41] and the small size of the population covered [54]. Among them, the smaller size seems to be the most important determinant—this is, at the same time, the consequence of the rural location and the main factor determining poor financial condition. Previous findings prove that a smaller hospital is heavily exposed to financial distress, regardless of the ownership or the aim of activity [51–54]. Bigger hospitals, despite their rural location, can achieve as good effectiveness as their urban counterparts [50]. So, to ensure the continuance of such economically sensitive hospitals [64], policymakers should employ a system of support [54], for example in the form of special rules of payment for the benefits that they provide [31,65]; this can also help to improve public perception of the organization [66].

Although the hospital's activity is very specific, however, in economic terms the enterprise and its financial condition strongly affect the quality of services [67]—higher direct costs are related to lower readmission rates [68]. Though hospitals in Poland are not-for-profit at the vast majority, they are obliged to keep the financial balance to continue their activity. This means there is a need to assess financial health on the basis of the indicators used in enterprises. Some of them are modified by introducing the specific values characterizing hospitals' activity, for example a number of beds. Usually, the following financial indicators are employed: (1) Profitability, (2) liquidity, (3) capital structure, (4) revenue indicators, (5) costs, and (6) utilization (bed occupancy) [57,68,69]. Most of the previous studies confirm rural hospitals' poor financial performance (based on indicators listed above) [37,69,70], while some results are inconsistent, suggesting that rural hospitals can be as profitable (or efficient) as their urban counterparts [58,71]. According to the pillars of the environment of the traditional definition of sustainable development, the efficiency of rural hospitals can help them use their resources in the best way and to direct them toward their missions of patient care. Due to the nature of the services they provide, health services use significant amounts of energy and water and generate large volumes of waste. Thus, their efficiency is fundamental.

## 4. Data

A major concern in the assessment of the financial conditions of rural hospitals is, in fact, the definition of rurality, which should take into account the specific character of a given country. In this study, we cannot directly adopt the definition proposed by the Polish Main Office, which defines rural areas, as "the areas situated outside the administrative boundaries of cities" (rural municipality, rural parts of the urban-rural municipalities) [72]. We cannot employ this definition in our study due to the fact that rural areas, based on this statistical spin, are strongly diversified and some of them have more urbanized character despite the low population density [73,74]. Due to the lack of a wider definition of a "rural hospital", we employ our own definition, assuming that a "rural hospital" fulfills contemporaneously the following criteria:

(1)　Is located in a county town;
(2)　The population of the county is lower than 100,000 people.

In practice, it means a rural area with one small urban center where a hospital is located. As a result, the hospital serves patients not only from the city center but also, and perhaps above all, from the surrounding villages; then, it can be assumed that the health service provider can be assessed as a "rural hospital". The definition of an urban hospital is similarly formulated. Only hospitals located in cities with a population exceeding 100,000 residents qualified for the analysis. Additionally, all hospitals located in "Katowice urban area" (highly urbanized part of Poland with population density above 1500/km$^2$) were qualified as urban hospitals. All hospitals owned by the regional authorities (NUTS 2), regardless of their real location, were included in the research sample as "urban hospitals". Such hospitals, due to their regional nature and more specialized services, usually serve very large populations.

The research data was collected by hand from the Emerging Markets Information Service [EMIS] Database, covering the years 2012–2017. Initially, we analyzed 1123 entities classified in the database

as "hospitals" or "hospital and medical activity". First, 327 entities were excluded due to lack of all the required data. During the next stage of data construction, based on the detailed analysis of every entity, we excluded:

(a)   Entities which provide mainly other services than stationary health care;

(b)   Hospitals, which provide primarily long-term care (psychiatric hospitals, rehabilitation hospitals, sanatoriums) because of the specificity of the activities;

(c)   Entities, providing mainly ambulatory care and "one-day procedures", classified in the database as "hospitals and medical activity";

(d)   Hospitals providing services in only one specialization (for example cardiology or radiology) due to its specificity.

"Rural hospitals" characterized by annual income higher than 22 million euro and providing medical services for the regional population regardless of their location in rural areas have also been excluded from the research sample. "Urban hospitals" with annual income lower than 6 million euro have also been rejected, in order to exclude small urban hospitals providing a low range of services, especially "one-day" surgical procedures. Ultimately, the research sample consisted of 150–199 hospitals, depending on the year (Table 1). Whenever the wording "rural hospital" or "urban hospital" appears in this paper, it refers to the definitions of "rural hospital" and "urban hospital" adopted in this study.

**Table 1.** Number of observations.

| Division | 2012 | 2013 | 2014 | 2015 | 2016 |
|----------|------|------|------|------|------|
| Rural    | 80   | 80   | 80   | 76   | 98   |
| Urban    | 70   | 70   | 70   | 85   | 101  |
| Total    | 150  | 150  | 150  | 161  | 199  |

## 5. Methodology

According to the literature review, we analyzed four pillars of financial health: Profitability, liquidity, efficiency, and debt using 7 variables (Table 2) which were selected based on their descriptive statistics. Profitability seems to be, in the light of previous studies, the most important factor determining a hospital's financial health. It can be measured using several financial indicators, which were selected due to their importance to the assessment of its financial condition in the case of Polish hospitals. For example, Polish hospitals usually pursue to obtain a positive value of EBIDTA (Earnings before Interest, Taxes, Depreciation, and Amortization), while EBIT (Earnings before Interest and Taxes) remains negative. Indicators like ROS (Return on Sales) or total margin, which are the best synthetic indicators of the profitability on sales [75], could not be employed in this study, because the hospitals analyzed operate in different organizational forms (public entities, companies owned by public bodies, private companies) [76]. Additionally, we checked the differences in size (both in terms of assets and income) between rural and urban hospitals.

The differences between the values of financial indicators for rural and urban hospitals were tested using the non-parametric Mann–Whitney U test due to the abnormal distribution of all analyzed ratios.

In the second stage of research, in order to provide a more comprehensive assessment of the financial health, we employed a synthetic measure of a hospital financial condition (M2) created based on the gradient method. The gradient was based on the determination of taxonomic distances between examined objects and defined reference points (bottom, top) [77–79] (see also, Appendix A). The obtained values range is 0–1 [80,81]. In order to build this synthetic measure, we used the selected indicators of profitability, liquidity, debt, and efficiency employed in the first stage of the study (Table 3) [82–85].

**Table 2.** Financial indicators employed in the research.

| Ratio | Formula | Character | Group |
|-------|---------|-----------|-------|
| OPM | EBIT/Sales | stimulant | profitability |
| CR | Current Assets/Current liabilities | nominant | liquidity |
| D% | Total debt/Total Assets | destimulant | debt |
| CF/Debt | (Net Profit + Depreciation)/Total debt | stimulant | debt |
| TAT | Sales/Total Assets | stimulant | efficiency |
| CES | Employee benefit expense/Sales | destimulant | efficiency |
| ROCF | (Net Profit + Depreciation)/Total Assets | stimulant | profitability |
| ASSETS | Ln Total Assets | nominant | size |
| INCOME | Ln Revenue from sales | stimulant | size |

Sales—the revenue from provided services, both from contracts with NFZ and private sources (only in the case of companies)

**Table 3.** Financial indicators chosen to construct the synthetic measure M2.

| Ratio | Formula | Character | Group |
|-------|---------|-----------|-------|
| OPM | EBIT/Sales | stimulant | profitability |
| CR | Current Assets/Current liabilities | nominant | liquidity |
| D% | Total debt/Total Assets | destimulant | debt |
| TAT | Sales/Total Assets | stimulant | efficiency |
| CES | Employee benefit expense/Sales | destimulant | efficiency |

Where: Nominants and destimulants have been converted into stimulants respectively: nominants: $x\_ij:=-|x\_ij-avarage(x\_i)|$, destimulants: $x\_ij:=[-x\_ij]$ [fo]

We defined the minimum and maximum values in the sample for the years 2012–2014 using the following formula:

$$M2 = 0.29196 * OPM - 0.031242 * CR - 0.84 - 0.031112 * D\% + 0.017609 * TAT - 0.066345 * CES - 0.75344 \quad (1)$$

During the last stage, we estimated three random-effects models with time dummies where the independent variables were: Legal form (0—public entity, 1—company), size (ln revenue), and localization (0—urban, 1—rural), and the dependent variable was the M2 indicator. We estimated models for the whole group and for rural and urban hospitals separately.

## 6. Results and Discussion

In this study, we form the assumption that hospitals located in rural areas are smaller and characterized by more difficult financial situations than hospitals in urban areas. The literature review indicates that the source of this disadvantage may be a smaller size, which enables hospitals to benefit from the economy of scale, and/or less specialized range of services. Employed measures of the size better than a number of beds reflect the potential to generate cash flows. Additionally, the value of total assets approximates the size of a hospital, not only in the physical dimension, but it also reflects the value of the hospital's equipment (which also affects the volume of revenue). The operating revenue is strictly associated with both the volume of provided services and its level of specialization, assuming that more specialized health benefits are better compensated. Our study confirms that rural hospitals in Poland are in fact smaller—both in terms of operating revenue and the value of total assets—and these differences are statistically significant at the level of $\alpha = 1\%$ in all analyzed years (Table 4). This is generally consistent with the characteristics presented in the literature which confirms the smaller size of rural hospitals [37,41,49,86].

**Table 4.** Results of Mann-Whitney U test—differences between urban and rural hospitals.

| 2012 | | | |
|---|---|---|---|
| **Ratio** | **Rural** | **Urban** | ***p*-value** |
| OPM | 6149 | 5176 | 0.6827 |
| CR | 6239 | 5086 | 0.4546 |
| D% | 5596 | 5729 | 0.0948 |
| CF/Debt | 6439 | 4886 | 0.1333 |
| TAT | 6239 | 5086 | 0.4546 |
| CES | 5718 | 5607 | 0.2258 |
| ROCF | 6146 | 5179 | 0.6911 |
| Ln(Assets) | 4610 | 6716 | 0.0000 |
| Ln(Sales) | 4363 | 6962 | 0.0000 |
| M2 | 6808 | 4517 | 0.0038 |

| 2013 | | | |
|---|---|---|---|
| **Ratio** | **Rural** | **Urban** | ***p*-value** |
| OPM | 5926 | 5399 | 0.4760 |
| CR | 6129 | 5196 | 0.9609 |
| D% | 5770 | 5555 | 0.1933 |
| CF/Debt | 6243 | 5082 | 0.6320 |
| TAT | 6539 | 4786 | 0.1107 |
| CES | 5725 | 5600 | 0.1414 |
| ROCF | 6021 | 5304 | 0.7230 |
| Ln(Assets) | 4529 | 6796 | 0.0000 |
| Ln(Sales) | 4435 | 6890 | 0.0000 |
| M2 | 6763 | 4562 | 0.0147 |

| 2014 | | | |
|---|---|---|---|
| **Ratio** | **Rural** | **Urban** | ***p*-value** |
| OPM | 5868 | 5457 | 0.5182 |
| CR | 5862 | 5463 | 0.5037 |
| D% | 5700 | 5625 | 0.2009 |
| CF/Debt | 6272 | 5053 | 0.3832 |
| TAT | 6443 | 4882 | 0.1295 |
| CES | 5812 | 5513 | 0.3914 |
| ROCF | 5980 | 5345 | 0.8226 |
| Ln(Assets) | 4493 | 6832 | 0.0000 |
| Ln(Sales) | 4398 | 6927 | 0.0000 |
| M2 | 6613 | 4712 | 0.0310 |

| 2015 | | | |
|---|---|---|---|
| **Ratio** | **Rural** | **Urban** | ***p*-value** |
| OPM | 6034 | 7007 | 0.6808 |
| CR | 6220 | 6821 | 0.8297 |
| D% | 6060 | 6981 | 0.7464 |
| CF/Debt | 5934 | 7107 | 0.4532 |
| TAT | 6727 | 6314 | 0.0534 |
| CES | 6290 | 6751 | 0.6512 |
| ROCF | 5945 | 7096 | 0.4760 |
| Ln(Assets) | 4502 | 8539 | 0.0000 |
| Ln(Sales) | 4416 | 8625 | 0.0000 |
| M2 | 6695 | 6346 | 0.0682 |

| 2016 | | | |
|---|---|---|---|
| **Ratio** | **Rural** | **Urban** | ***p*-value** |
| OPM | 9659 | 10,241 | 0.7294 |
| CR | 9882 | 10,018 | 0.8410 |
| D% | 9673 | 10,227 | 0.7555 |
| CF/Debt | 9601 | 10,299 | 0.6250 |
| TAT | 10,695 | 9205 | 0.0276 |
| CES | 9863 | 10,037 | 0.8777 |
| ROCF | 9686 | 10,214 | 0.7799 |
| Ln(Assets) | 7387 | 12,513 | 0.0000 |
| Ln(Sales) | 7276 | 12,624 | 0.0000 |
| M2 | 10,975 | 8925 | 0.0038 |

This difference should significantly influence the financial health of a hospital. Higher income means a higher scale of activity, but it can be also related to a higher intensity of care as a result of providing more specialized services. Regardless of the source of this difference, it may be associated with the ability to exploit economies of scale—the median value of assets for urban hospitals is EUR 11.45 million, whereas for rural hospitals it is EUR 4.53 million. The volume of assets translates partially into the ability to generate revenue—the median of revenue for hospitals located in urban areas is EUR 13.57 million whereas for rural hospitals it is EUR 6.38 million (data for 2016).

The main part of the research, which relates to the H1 hypothesis, consists of testing differences in the financial health for urban and rural hospitals. We cannot confirm the difference at the level of the individual indicators—these differences are not statistically significant—however, if we apply the synthetic measure of financial condition (M2), we can observe that the overall financial condition of rural hospitals is better than urban of the urban ones in every analyzed year and those differences are statistically significant for all years (except 2015) (Table 4).

The estimated models (Table 5) confirm that rural hospitals are characterized by better financial condition (the value of the variable "rural" is positive (0.023) and statistically significant). We also find that hospitals with higher revenue have better financial health (the variable "Ln Revenue" has a positive coefficient (0.015) and is statistically significant). Also, hospitals operating in the form of companies achieve better financial condition than those operating in the form of SPZO—the variable "Legal_form" has a positive coefficient (0.030) and is statistically significant. The same relationship can be observed at the level of the whole sample and in the case of urban hospitals, but the increase in revenue boost the financial condition of urban hospitals is slightly stronger than in the case of the whole sample.

**Table 5.** Regression results.

| | Whole Sample | | Urban Only | | Rural Only | |
|---|---|---|---|---|---|---|
| | **Coefficient** | ***p*-Value** | **Coefficient** | ***p*-Value** | **Coefficient** | ***p*-Value** |
| Const | 0.53791 (0.05824) | <0.0001 | 0.45424 (0.07920) | <0.0001 | 0.70449 (0.03730) | <0.0001 |
| Legal_form | 0.03073 (0.00503) | <0.0001 | 0.04280 (0.00763) | <0.0001 | 0.02180 (0.00541) | <0.0001 |
| Ln Revenue | 0.01571 (0.00588) | 0.0076 | 0.02376 (0.00806) | 0.0032 | 0.00034 (0.00406) | 0.9315 |
| dt_2 | −0.00099 (0.00218) | 0.65 | −0.00269 (0.00389) | 0.4894 | −0.00032 (0.00163) | 0.8424 |
| dt_3 | −0.00140 (0.00228) | 0.5382 | −0.00254 (0.00365) | 0.4868 | −0.00161 (0.00244) | 0.5084 |
| dt_4 | −0.02517 (0.00904) | 0.0054 | −0.03622 (0.01338) | 0.0068 | −0.00656 (0.00636) | 0.3019 |
| dt_5 | −0.02770 (0.00932) | 0.003 | −0.03935 (0.01362) | 0.0039 | −0.00741 (0.00673) | 0.2706 |
| Rural | 0.0238431 (0.00673) | 0.0004 | | | | |
| Mean dependent variable | 0.708122 | | 0.69962 | | 0.716214 | |
| Sum squared residuals | 1.010879 | | 0.625202 | | 0.35885 | |
| Log-likelihood | 1558.576 | | 713.1112 | | 874.6649 | |
| Schwarz criterion | −3063.576 | | −1384.370 | | −1707.132 | |
| Rho | −0.271669 | | −0.313066 | | −0.216595 | |
| S.D. dependent variable | 0.040083 | | 0.045105 | | 0.032693 | |
| S.E. of regression | 0.035481 | | 0.04009 | | 0.029621 | |
| Akaike criterion | −3101.153 | | −1412.222 | | −1735.330 | |
| Hannan–Quinn | −3086.726 | | −1401.187 | | −1724.179 | |
| Durbin–Watson | 1.524023 | | 1.627987 | | 1.421658 | |

When we analyze the model estimated only for rural hospitals, a completely different pattern can be observed. In the case of these hospitals, only the legal form of activity is statistically significant. A positive coefficient (0.02) for the variable "Legal form" indicates that the form of a company has a positive effect on the financial health, although to a lesser extent than in the case of urban hospitals (0.04) or at the level of the whole sample (0.03). In the case of rural hospitals, the size does not matter—an increase in revenue does not improve the financial health.

Our hypothesis was constructed on the basis of a literature review. Surprisingly, the presented results do not expressly support this assumption. The analysis of financial indicators clearly shows that there are no statistically significant differences between rural and urban hospitals, or even that the financial health of rural hospitals is better when we employ the synthetic measure. These results are inconsistent with those obtained in previous studies [41,49,50,63,76,86,87].

It seems that a smaller size, which in previous studies was seen as a factor increasing the risk of financial distress [28,34,86], might be even a source of competitive advantage. Large hospitals located in cities usually have higher assets, which, apart from generating higher revenues, may be associated with higher costs. Another risk factor indicated in the literature is a smaller range of benefits [50,63,76,87,88]. A smaller range of benefits usually means less specialized procedures. Rural hospitals usually do not provide highly specialized life-saving procedures (invasive cardiology, transplants). Such highly specialized procedures are a source of higher income per bed, provided that they are well valued by the payer. Otherwise, they can lead to the deterioration of the financial situation.

The study identifies other important drivers of rural hospitals' financial health. We prove that the form of activity is very important, while it forces the hospital to maintain financial health. On the other hand, rural hospitals cannot improve their situation by seeking to increase the scale of medical activity, which can be a source of improvement in the case of urban entities.

## 7. Conclusions

We studied the economic situation of rural hospitals due to its importance for local communities. For the majority of the inhabitants of rural areas, the existence of a hospital in short means better access to benefits. Patients have a chance to maintain close contact with their families, which is especially important in the case of children and the elderly. However, the hospital is also an enterprise, though a very specific one. To be able to survive, it must keep financial condition sufficient to finance its current medical activity. The literature review presented in this research proves that rural hospitals are smaller and economically more sensitive, but those research results come generally from the American market. European studies on the financial health of rural hospitals are very few and, in the case of Poland, this is the first such study. Therefore, our results represent a very significant contribution to the science. Although our work is based on a relatively small research sample, we can conclude that rural hospitals, though smaller both in terms of income and assets, are at a lower risk of financial distress than their larger urban counterparts.

The result suggest that the size is not the main determinant of hospital financial performance. In the case of hospitals, the form of activity seems to play a crucial role—entities operating in the form of a company are forced to keep a greater financial discipline. This also explains the process of transformations into companies which can be observed in recent years. Paradoxically, the increasing scale of operations (increase in revenue) does not improve financial health—hospitals located in cities benefit more from such processes (the consolidation of hospitals). On the other hand, the fact that the level of revenue of rural hospitals does not affect the financial health suggests greater resistance to changes in the external environment.

This research is an important contribution to the discussion on the role and financial condition of rural hospitals. In all European countries, there are hospitals located in remote areas, distant from the large urban centers. As we have demonstrated, hospitals located in smaller centers, although smaller in terms of revenue and assets, are not characterized by a worse financial situation. Then, they are in line with the economic and environmental pillars of the sustainable development. Rural hospitals can use their economic and moral positions within their communities to help them achieve the two objectives of sustainable development goals related to health as well as sustainability and foster green economy. We can even hypothesize that smaller size is the source of their competitive advantage, just like the fact that they often operate as commercial law companies. It seems that our results might determine the new directions of changes for other European countries—creating networks of small,

flexible hospital units, responsive to the needs of local communities, capable of providing equal and effective access to health services, in line with the social pillar and respect of rural communities.

The better, or just sufficient, financial condition of rural hospitals supports the concept of rural sustainability on many levels (Figure 1). The rural sustainability is focused on the local economic, social, and environmental development to create conditions for the elimination of poverty and better quality of life.

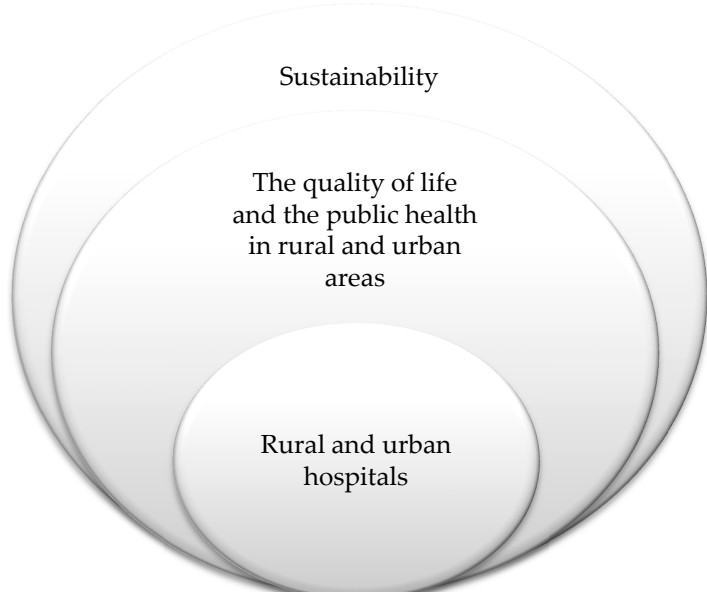

**Figure 1.** Impact of rural and urban hospitals on society and environment. Authors own elaboration.

The environmental protection and mitigation of climate change support the sustainable development of agriculture and better quality of food supply, which is important for public health. The rural health infrastructure and access to basic health services and education create opportunities for rural areas to develop, and it is critical for sustainability and national wellbeing. The general trend in the economy is orientated towards decentralization of production, which means also decentralization and democratization of social activities. Also, urban and local development is playing an important role in the decentralization process.

Sustainable rural development is also an answer to the crisis of conventional, industrial agriculture which is changing rural areas. A broader spectrum of risks emerged, especially health risks. The factors to increase health risk are environmental damage, pollution, soil erosion, and low quality of food. Redefinition of socio-environmental balance is necessary for better quality of living in the rural and urban areas. The good financial condition of rural hospitals can result in:

(1) Better access to health services;
(2) Reduction of health risks (climate, environmental, disabilities, aging);
(3) Elimination of poverty by dedicated actions for vulnerable households, in particular, the aged, persons with disabilities and the unemployed;
(4) Reducing costs of health services for citizens (better access, transport exclusion);
(5) According to the sustainable development goals, it can encourage rural communities, increase their participation in decision-making, and empower rural leadership;
(6) Reinforce environmental and social resilience in rural areas;
(7) As a result, public health and sustainability issues go together and try to establish a framework to face many challenges in rural and urban areas. The crucial part of the new approach is its long-term perspective, responsibility, and complexity. Public health must be aligned with sustainable development and climate change.

The primary weakness of the study is the relatively small research sample. We also adopt our own definition of "rurality", which can influence the obtained results. In further research, we plan to verify whether the change in criteria affects the results obtained.

**Author Contributions:** Conceptualization, A.B. and P.G.; methodology, R.S.; validation, A.B. and P.U.-J.; formal analysis, P.P.; investigation, A.B. and P.P.; data curation, R.S.; writing—original draft preparation, A.B.; writing—review and editing, P.U.-J.; supervision, B.R.

**Funding:** This research received no external funding.

**Conflicts of Interest:** The authors certify that they have no involvement in any organization or entity with any financial interest or non-financial interest in the subject matter or materials discussed in this manuscript.

**Appendix A**

A gradient method is a taxonomic tool based on determination of taxonomic distances of the examined objects from defined reference points [78,81]. This procedure allows construction of a synthetic indicator of different nature, by combining values of variables denominated in different units, including dummy ones. Variables might be of a financial and non-financial character but must be stimulant—nominant and destimulant variables should be transformed into stimulant ones.

The method assumes that the matrix X comprises financial ratio values (observations of the studied phenomenon) denoted as:xij, which can be converted into stimulants (Destimulants and nominants have to be converted into stimulants) $x_{ij}$ where

i = 1,2,3, . . . , m, (a number of analyzed indicators—financial ratios);
j = 1,2,3, . . . , n, (a number of analyzed observations—hospitals);
and $x_{ij} \in$ R.
In order to measure a taxonomic distance, two points must be determined:
Top: $P = [p_1, p_2, p_3, \ldots, p_m]$
Bottom: $Q = [q_1, q_2, q_3, \ldots, q_m]$
where $p_i = \max_j x_{i,j}$ and $q_i = \min_j x_{i,j}$

As the $QP$ segment describes the axis of synthetic indicator, the $PQ$ vector gradient takes a form of linear programming function:
$\Phi(X) = [P - Q]X^T$ and values of this function represent the value of the synthetic indicator, according to the formula:

$$\varphi = (p_i - q_i) * x_{i,j} \tag{A1}$$

The obtained values of specific indicators, due to their construction, might take potentially very dissimilar values. In this situation, some indicators would affect a synthetic measure more strongly than others. To avoid this effect, the obtained values are reduced to the range of 0–1, using the scaling method. Conversions should be made from matrix X to Z according to the following formula:

$$for\ every\ i\ z_{ij} = \frac{x_{ij} - \min(x_{ij})}{\max(x_{ij}) - \min(x_{ij})} \tag{A2}$$

As a result, points P and Q take the following form: $P = |1, \ldots 1|, Q = |0, \ldots 0|$:

$$\varphi = \sum_{i=1}^{m} z_{ij} \tag{A3}$$

and the measure of development M (M1 and M2) is defined as:

$$M = \frac{\varphi}{m} \tag{A4}$$

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
