# Peer review of "Hospitals’ Financial Health in Rural and Urban Areas in Poland: Does It Ensure Sustainability?"

_sustainability, doi:10.3390/su11071932_

Round 1

Reviewer 1 Report

Thank you for the opportunity to review your paper about the financial performance of rural hospital in Poland and the related impact on sustainability.

The are many weaknesses to be addressed.

Firstly the analysis of the relation between rural hospitals and sustainable development should be improved; the related link is not clear and it should be better explained.

The literature review should be improved as well

Before analysing rural hospitals, I suggest to explain the main features of Poland Health System and to make a clear distinction between public and private hospitals as they operate in a different manner.

The implications of research are not clearly stated.

Methodology and experiments present many flaws

I hope you find these comments helpful for furthering your work.

Author Response

Anonymous Referee #1

C1 & C2: The analysis of the relation between rural hospitals and sustainable development should be improved; the related link is not clear and it should be better explained. The literature review should be improved as well

A1 & A2: We have extended the literature review in the area of rural hospital’s financial performance as well as in the problem of the role of hospitals as a part of sustainable rural development. Finally, we analysed 33 additional publications.

C3: Before analysing rural hospitals, I suggest to explain the main features of Poland Health System and to make a clear distinction between public and private hospitals as they operate in a different manner.

A3: We have added a new section (section 2) devoted to the problem of health care system in Poland, especially in the context of hospitals’ activity.

C4: The implications of research are not clearly stated.

A4: We have extended the section “conclusions” in order to present the implications in more comprehensive way.

C5: Methodology and experiments present many flaws.

A5: We have added the regression analysis (random-effects panel data) creating 3 models – for the whole group, and separately for rural and urban hospitals. Additionally we test the differences between values of rural and urban hospitals by employing the Mann-Whitney test.

Reviewer 2 Report

Please find attached the review report.

Author Response

Anonymous Referee #2

C1: The introductory section should reveal the novelty/originality of the paper.

A1: We emphasised the novelty of the study, but in the section “Conclusion”. This is definitely the first study in this area in Poland, while in the whole Europe those kind of study are very scarce – majority of them come from the American market.

C2: More facts towards the specifics of Polish health system are required.

A2: We have added the section with a brief description of the health care system in Poland, especially from the point of view of the hospitals.

C3: The second section should be extended with more previous related studies.

A3: We have done our best to extend literature review. Finally we added 33 publications presenting the problem of rural hospitals’ financial condition as well as the role of rural hospitals in building the sustainable rural environment.

C4: The third section should be merged with the secondary section.

A4: We have merged those parts.

C5: Even if the sample consists of 150 hospitals, the period of research is too short, namely three years (2012-2014), as well as a bit outdated. Hence, the sample should be updated.

A5: We have included the data from the year 2015-2016 into the analysis. The number of observations from the year 2017 was too small for the statistical analysis.

C6: I suggest the author(s) to comprise more specific variables (e.g. the number of hospital’ beds, the level of employment of skilled medical staff, remuneration of medical staff) in addition to financial indicators.

A6: We are aware that those specific variables would significantly enrich this analysis. Unfortunately, the data in this area, although they are collected by the public statistics, are not available even for scientific research.Some of the data, due to the nature of the hospital’s activity is untrustworthy. “The expenditure on salaries” is a good example - most doctors work on the basis of contracts for the provision of services, which are classified as "external services" (and there is also a variety of other services, such as cleaning or catering in this group).

C7: The author(s) should employ at least a regression analysis (e.g. fixed-effects or random-effects panel data regression models).

A7: We have added the regression analysis (random-effects panel data) creating 3 models – for the whole group, and separately for rural and urban hospitals. Additionally we test the differences between values of rural and urban hospitals by employing the Mann-Whitney test.

Reviewer 3 Report

I have read the whole paper that is written in the area of Poland’s health care. My understanding of the paper is given below:

1.       The research deals with hospitals’ financial performance. But, how it is associated with sustainability I am in doubt. Sustainability is just not a word, it has a clear conceptual framework. The paper fails to discuss the sustainability concept.

2.       What is the research question and objectives? There is no logical explanation of the research questions, objectives and implications.

3.       The performance of a hospital is measured by services. Financially sound doesn’t ensure the soundness quality of a hospital because there are two types of hospital private and public. The private hospital aims to earn more money than the public. But, the paper deals only rural and urban. The given classifications have also private and public hospitals but they didn’t mention its performance.

4.       The study is for the period of 2012-2014. The current year is 2019, it is unexpected to conduct research based on 5 years prior data. 2015 and 2016 data has some problems therefore, what about 2017 and 2018? Moreover, how do we believe 2012-2014 data?

5.       There is a hypothesis but how the hypothesis is driven there is no sound and logical theoretical understanding. Moreover, the literature behind the hypothesis is scattered and vague.

6.       In the different table there is some measurement and indicators of financial variables but unfortunately, there is no variable description, full meaning and the reason behind using these variables.

7.       Hospitals is a service industry and of course, they have profits but the profit measurement techniques are not similar to the merchandising or trading industry. What are the EBIT, sales……not clear to me because the term sales are irrelevant to the service organization like a hospital? Moreover, profitability indicator lies many variables that are above the profitability concepts. 

8.       Size is a big matter. Therefore, big size indicates more profitability it doesn’t true all time because profitability depends on the nature and motive of the business. Moreover, there are many specialized hospitals even in urban and rural areas and they have different motives.

9.       Figure 5….how does it from? It looks very ridiculous that the graph shows the relationship of rural hospitals sustainability. So, what about the urban hospital sustainability?

10.   What is the applications and significance of the study?

11. The sustainability concept must be revised. 

I hope the author(s) would seriously revisit the above issues and consider a sound question, objective, theory and diverse literature. 

Author Response

Anonymous Referee #3

C1: The research deals with hospitals’ financial performance. But, how it is associated with sustainability I am in doubt. Sustainability is just not a word, it has a clear conceptual framework. The paper fails to discuss the sustainability concept.

A1: In order to explain the link between sustainability and rural hospitals’ financial performance in the more comprehensive way we have extended section 3 “Rural hospitals as part of a sustainable health system” as well as the “Conclusions” section. We try to show that the rural hospitals continued existence is crucial from the point of view of the sustainable rural environment (as a part of social infrastructure), while it is impossible without a stable financial condition.

C2: What is the research question and objectives? There is no logical explanation of the research questions, objectives and implications.

A2: We have extended the literature review to show the research questions and objectives in the wider concept. We have also improved the implications in the last section.

C3: The performance of a hospital is measured by services. Financially sound doesn’t ensure the soundness quality of a hospital because there are two types of hospital private and public. The private hospital aims to earn more money than the public one. But, the paper deals only rural and urban. The given classifications have also private and public hospitals but they didn’t mention their performance.

A3: The paper deals with the problem of rural hospitals’ financial condition (comparing to the urban ones) as a key factor which allows to continue their activity, which built an equal access to health benefits in rural areas. From the point of view of the Polish hospitals the distinction between private and public hospitals has a secondary character. Private hospitals are really few and usually they provide a very narrow range of services (for example only procedures in ophthalmology). Such hospitals with only one specialisation were excluded from the sample in order to keep it more homogenous. From the point of view of Polish hospitals the form of activity is much more important (a company or SPZOZ), but the form of a company does not imply the for-profit orientation (those hospitals are usually still public). We have also added a section with a brief description of the Polish health care system to make our dilatations more clear.

C4: The study is for the period of 2012-2014. The current year is 2019, it is unexpected to conduct research based on 5 years prior data. 2015 and 2016 data has some problems therefore, what about 2017 and 2018? Moreover, how do we believe 2012-2014 data.

A4: We have included in our analysis the data for the years 2015 and 2016 which are now available.

C5 & C6: There is a hypothesis but how the hypothesis is driven there is no sound and logical theoretical understanding. Moreover, the literature behind the hypothesis is scattered and vague. In different tables, there are some measurements and indicators of financial variables but unfortunately, there is no variable description, full meaning and the reason behind using these variables.

A5: We have added 33 publications to the literature review, which also helps us to explain better the research analysis and the reasons to use the chosen financial indicators.

C7: Hospitals is a service industry and of course, they have profits but the profit measurement techniques are not similar to the merchandising or trading industry. What are the EBIT, sales……not clear to me because the term sales are irrelevant to the service organization like a hospital? Moreover, profitability indicator lies many variables that are above the profitability concepts.

A7: We agree with the Reviewer 2 that the concept of profitability may be unclear in the case of non-profit hospitals, although the literature review suggests that we should use the same indicators in the case of hospitals as in the case of other companies. There is much evidence in this area, especially taken from the American market. Of course, we agree that the concept of “sales” or EBIT in the case of hospitals can be confusing – according to that we have added the definitions in the Table 3.

C8: Size is a big matter. Therefore, big size indicates more profitability it is not true all the time because profitability depends on the nature and motive of the business. Moreover, there are many specialized hospitals even in urban and rural areas and they have different motives.

A8: We fully agree that big size doesn’t imply higher profitability. In this study that rural hospitals, although smaller, in terms of their assets or revenue, can perform as well as bigger urban entities, or even better as suggested by a synthetic measure of financial condition.

C9: Figure 5….how does it from? It looks very ridiculous that the graph shows the relationship of rural hospitals sustainability. So, what about the urban hospital sustainability?

A9: This Figure is now corrected

C10: What is the applications and significance of the study?

A10: We have corrected the section “Conclusions” in order to stress the potential applications and significance of the study.

C11: The sustainability concept must be revised.

A11: We have revised the concept of sustainability by introducing the extended literature review.

Round 2

Reviewer 1 Report

The paper has been improved; although, even if the authors have tried to explian the link between rural hospitals and sustainability, I think that there is not a clear relation between the paper’s topic and the Sustainability Journal domain. In your paper, the relation between sustainability – identified according to the traditional definition - and rural hospitals looks like a bit of a stretch ; in your paper sustainability mainly refers to Poland rural health system’s financial conditions rather than the concurrent achievement of economic, social and environmental results.

Author Response

Anonymous Referee #1

C1: In your paper, the relation between sustainability – identified according to the traditional definition - and rural hospitals looks like a bit of a stretch ; in your paper sustainability mainly refers to Poland rural health system’s financial conditions rather than the concurrent achievement of economic, social and environmental results.

A1: In our study we try to refer to the Special Issue’s title: "Social Public Health System and Sustainability".  We agree that the relationship between the rural hospitals’ financial condition and rural sustainable development may not be obvious at first glance. We can, however, accept the assumption that equal access to health services is one of the elements that build sustainable social environment. Evidence based on the literature shows that in rural areas a network of healthcare providers is scarcer - this also applies to primary care and specialist outpatient care. As a result, rural residents are less likely to use outpatient services while they are forced to use emergency medicine more extensively. Hence, the existence of hospitals in rural areas provides a better access to benefits. A hospital cannot function if its financial situation is unstable. In the literature review we show that in many countries, rural hospitals are in a difficult financial situation, which threatens their survival. However, the results of our research indicate that rural hospitals in Poland are not in a worse financial condition than their urban counterparts. This indicates that, so far, there has been no need to employ any special funding rules for such hospitals (contrary to the solutions applied in other countries).

Reviewer 2 Report

The revised version of the manuscript entitled “Hospitals’ Financial Health in Rural and Urban Areas in Poland. Does It Ensure Sustainability?” (Manuscript ID: sustainability-456510) improved noticeably. The author(s) incorporated the suggested changes and provided proper arguments and suitable explanations. Therefore, the paper is worth publishing. Henceforward, I recommend paper acceptance.

Author Response

Anonymous Referee #2

C1: The author(s) incorporated the suggested changes and provided proper arguments and suitable explanations. Therefore, the paper is worth publishing. Henceforward, I recommend paper acceptance.

A1: We’d like to thank a lot to the reviewer for this positive feedback, which we appreciate. Thanks to the valuable comments the readers are likely to find our paper new and interesting.

Reviewer 3 Report

I appreciate the author(s) hard work. They worked a lot to improve the paper. Now the paper looks better than the prior version but I am still in the same mode because of the rigorousness, conceptual framework, and the applications.

Another major concern is about data. In the prior version, they mentioned about the problems of the data of 2015 and 2016 (we were unable to collect the data for the years 2015 and 2016 due to the numerous missing data-points). But, in the version, they used the same data size (2015) and more in (2016). It raises the question and douts of the research. 

Author Response

Anonymous Referee #3

C1: I appreciate the author(s) hard work. They worked a lot to improve the paper. Now the paper looks better than the prior version but I am still in the same mode because of the rigorousness, conceptual framework, and the applications.

A2: We are, once again, grateful for all comments which allowed us to improve the paper. We try to explain the application of our research in the section “Conclusions”.

C2: Another major concern is about data. In the prior version, they mentioned about the problems of the data of 2015 and 2016 (we were unable to collect the data for the years 2015 and 2016 due to the numerous missing data-points). But, in the version, they used the same data size (2015) and more in (2016). It raises the question and doubts of the research.

A2: We started the first study in this area in the year 2016. The first version of this paper was developed in the year 2017 based on database built at the beginning of 2017– at that time, the statistics from the year 2016 year were not available and the statistics from the year 2015 were in fact characterised by the numerous missing data-points.When preparing the paper for “Sustainability”we focused on matters related to sustainable development of rural areas and we extended the literature review. We admit that submitting the paper without updated calculations was an inexcusable negligence. Guided by the reviewers’ suggestions, which was justified, we loaded again the data for the period 2015 - 2016 (updated in the EMIS Database ). In addition, we have provide the database on the basis of which we have performed calculations in order to validate the obtained results.

Round 3

Reviewer 1 Report

I understand your explanation about the relationship between sustainability and rural hospital. Anyway, I recommend you to better explain this relation in your paper, according to a traditional sustainability definition

Author Response

Anonymous Referee #1

C1: I understand your explanation about the relationship between sustainability and rural hospital. Anyway, I recommend you to better explain this relation in your paper, according to a traditional sustainability definition

A1: Based on the traditional definition of sustainable development proposed by the Burtland World Commission on Environment and Development in 1987, in the paragraphs number 17 and 18 we explain the impact of rural hospitals on three pillars of sustainable development (lines 216-237), also by extending the literature review. We also introduce additional corrections / clarifications according to those pillars of sustainable development in the lines: 140, 175-178, 190-193, 196-198, 203, 211-215, 256-257, 268-272, 431-435, 439-440.

Reviewer 3 Report

Appreciate the author(s) hard working.  

Author Response

Anonymous Referee #3

C1: Appreciate the author(s) hard working. 

A2: We are, once again, grateful for all comments which allowed us to improve the paper.